# ~~Interpretability-driven~~ Explainability-driven Active Feature Acquisition in Learning Systems

## Abstract

In real-world applications like medicine, machine learning models must often work with a limited number of features due to the high cost and time required to acquire all relevant data. While several static feature selection methods exist, they are suboptimal due to their inability to adapt to varying feature importance across different instances. A more flexible approach is active feature acquisition (AFA), which dynamically selects features based on their relevance for each individual case. Here, we introduce an AFA framework that leverages ~~Shapley Additive explanations~~ SHapley Additive exPlanations (SHAP) to generate instance-specific feature importance rankings. By reframing the AFA problem as a feature prediction task, we propose a policy network based on a decision transformer architecture, trained to predict the next most informative feature based on SHAP values. This method allows us to sequentially acquire features in order of their predictive significance, resulting in more efficient feature selection and acquisition. Extensive experiments across multiple datasets show that our approach achieves superior performance compared to current state-of-the-art AFA techniques, both in terms of predictive accuracy and feature acquisition efficiency. These results demonstrate the potential of ~~SHAP-based~~ explainability-driven AFA for applications where feature acquisition cost is a critical consideration~~, such as in disease diagnosis~~.

## 1 Introduction

In traditional machine learning settings, it is typically assumed to have all features available during inference. However, in real-world scenarios, ~~especially in medical settings,~~ acquiring these features can be expensive, time-consuming, and is often done sequentially. Therefore, it is crucial to develop methods that can make accurate predictions with a limited number of features. This can be achieved by selecting a static global subset of features, but it is suboptimal since the important set of features may vary across different instances (Kachuee et al., 2019; Covert et al., 2023b). Additionally, the chosen subset might not provide sufficient information for some cases, necessitating the acquisition of more features to ensure a confident prediction. A more effective strategy is to identify important features sequentially for each individual instance, a technique known as active (or dynamic) feature acquisition (AFA), which has been gaining increasing attention in recent years (He & Chen, 2022; von Kleist et al., 2023; Chattopadhyay et al., 2024).

The literature mainly contains two different ways of approaching AFA: reinforcement learning (RL)-based and greedy-based methods. Both approaches aim to develop a feature selection policy through exploration. RL-based methods (Kachuee et al., 2019; Yin et al., 2020; von Kleist et al., 2023) train policy networks by maximizing different reward functions. While the RL-based approach is intuitive for this sequential task and theoretically capable of finding the optimal policy, empirical evidence shows that RL-based methods often underperform compared to greedy-based methods ~~Gadgil et al. (2024)~~ (Gadgil et al., 2024). Greedy-based methods attempt to predict the next most important available feature by calculating conditional mutual information (CMI). To compute CMI, researchers have proposed both generative approaches (Rangrej & Clark, 2021; He et al., 2022) and methods based on the variational perspective (Covert et al., 2023b; Gadgil et al., 2024). However,

calculating CMI directly remains challenging, and methods leveraging the variational perspective have demonstrated superior performance compared to generative alternatives.

In this work, we approached the problem by empirically observing that deep ~~learning model~~ learning-based local explanation methods, such as ~~Shapley Additive Explanations~~ SHapley Additive exPlanations (SHAP) (Lundberg & Lee, 2017), can be utilized to identify instance-wise feature importance rankings. With this insight, we treated the AFA problem as a feature prediction task rather than a feature exploration one. Our contributions are listed below:

- To the best of our knowledge, this is the first time in the AFA literature that the utility of local explanation methods, specifically SHAP (Lundberg & Lee, 2017), has been demonstrated for determining instance-wise feature importance rankings. We demonstrate that if we had an ideal (oracle) policy network that sequentially selects features based on their SHAP values, sorted from highest to lowest during inference, would outperform current state-of-the-art AFA techniques in terms of accuracy for any fixed number of features. Similar observations were made in the local explanation literature (Petsiuk et al., 2018; Jethani et al., 2021; 2022). They illustrate that insertion (or deletion) of the important features ranked based on their respective explanation methods improves (or degrades) model performance. However, these observations have yet to be formally compared with AFA techniques, leaving a gap in understanding how AFA methods compare to these explanation-based feature ranking approaches.
- We ~~took a different approach by training~~ trained our policy network to predict the next unacquired feature with the highest SHAP value, based on the current observation.
- We employed recently developed decision transformer (Chen et al., 2021) architecture as a policy network, and trained it using a two-stage approach. We showed that the feature importance ranking order is predictable without observing ~~them~~ it. Also, our experiments demonstrate that our technique achieves better or comparable results with the state-of-the-art AFA methods on different datasets.

## 2    RELATED WORKS

Generally, the methods in the AFA literature have two networks: a policy network for feature acquisition and a prediction network for prediction with available subset of features. These methods mainly differ in training their policy networks, so we only highlight those differences.

The AFA problem can be formulated as a Markov decision process (MDP) (Zubek & Dietterich, 2002; Dulac-Arnold et al., 2011); based on this formulation, there have been many RL-based approaches proposed (Dulac-Arnold et al., 2011; Shim et al., 2018; Kachuee et al., 2019; Yin et al., 2020; Li & Oliva, 2021; von Kleist et al., 2023). These methods generally train their policy networks with the objective of maximizing the defined reward functions. Namely, they try to approximate the action-value function (i.e., Q-function). For example, in (Dulac-Arnold et al., 2011), the Q-function is approximated linearly and later it is extended in (Janisch et al., 2019) using a deep Q network (Mnih et al., 2015; van Hasselt et al., 2016). A similar approach was taken by the opportunistic learning (OL) method in (Kachuee et al., 2019). Another type of mainstream methods (Rangrej & Clark, 2021; He et al., 2022; Covert et al., 2023b; Chattopadhyay et al., 2023; Gadgil et al., 2024) are the greedy-based ~~methods~~ frameworks. These methods acquire the features by estimating the conditional mutual information (CMI) between the current available subset of features and the unacquired features. For CMI estimation, there are generative approaches (Rangrej & Clark, 2021; He et al., 2022) that use variational autoencoders (Kingma & Welling, 2013), and discriminative approaches (Covert et al., 2023b; Chattopadhyay et al., 2023; Gadgil et al., 2024) directly predicting the feature index with the highest CMI without explicitly calculating CMI. Although, the MDP formulation is theoretically appealing, RL-based methods often underperform compared to the discriminative approaches such as the greedy-based methods ~~Covert et al. (2023b); Gadgil et al. (2024)~~ (Covert et al., 2023b; Gadgil et al., 2024).

In addition to AFA methods, related approaches from the budget learning literature ~~Trapeznikov & Saligrama (2013); Nan & Saligrama (2017); Ekanayake & Zois (2024)~~ (Trapeznikov & Saligrama, 2013; Nan & Saligrama, 2017; Ekanayake & Zois, 2024) explore fixed feature acquisition orders, limiting the number of potential feature subsets. These methods aim to identify easily classifiable instances, enabling the acquisition of a minimal set of features in such cases, thereby reducing overall acquisition costs.

With regards to the local explanation literature (Petsiuk et al., 2018; Jethani et al., 2021; Lundberg & Lee, 2017), various methods focus on quantifying the contribution of individual features to model predictions for each instance. Among these methods, SHAP (Lundberg & Lee, 2017), based on game-theoretic Shapley values ~~Shapley (1953)~~(Shapley, 1953), is particularly popular. However, SHAP calculations are computationally intensive, leading to the development of several approximations (Lundberg & Lee, 2017; Ancona et al., 2019; Jethani et al., 2022; Covert et al., 2023a). FastSHAP (Jethani et al., 2022), for instance, provides an efficient approximation using a deep explainer model. Additionally, global feature importance methods aim to identify the most relevant static features across an entire dataset. For example, the Concrete Autoencoder (CAE) (Balın et al., 2019) trains an autoencoder to select important features, while SAGE (Covert et al., 2020) extends Shapley values to quantify global feature importance through an additive importance measure. For a detailed overview, we refer readers to recent surveys (Samek et al., 2021; Bolón-Canedo et al., 2022).

| Notation | Description |
|---|---|
| $\mathbf{x}$ | Input feature vector |
| $d$ | Dimensions of input feature vector |
| $y$ | Target label |
| $C$ | Number of classes |
| $q_\pi$ | Policy network~~(Causal transformer)~~ |
| $f_\theta$ | Predictor network |
| $\pi$ | Parameters of policy network |
| $\theta$ | Parameters of predictor network |
| ~~$\mathbf{r_t}$~~ $r_t$ | ~~Index of the latest acquired feature (reward~~Logits of the predictor (action) after acquiring $t$ features |
| $a_t$ | ~~Logits of the predictor (action~~Index of the latest acquired feature (reward) after acquiring $t$ features |
| $\hat{\mathbf{q}}$ | Output of policy network $q_\pi$ |
| $\hat{\mathbf{y}}$ | Output of predictor network $f_\theta$ |
| $\varphi^i(t)$ | $t^{th}$ important feature of $\mathbf{x}^{(\mathbf{i})}$ based on SHAP ranking |
| $\hat{\varphi^i}(t)$ | $t^{th}$ important feature of $\mathbf{x}^{(\mathbf{i})}$ based on policy network's predictions |
| $M_t$ | Set of $t$ most important feature indices based on SHAP ranking |
| $\hat{M}_t$ | Set of first $t$ feature indices acquired based on $\hat{\varphi^i}$ |
| $\mathbf{x}_{M_t}$ | Input feature vector with features from $M_t$ unmasked |
| $\mathbf{x}_{\hat{M}_t}$ | Input feature vector with features from $\hat{M}_t$ unmasked |
| $\ell$ | Context length of causal transformer |

Table 1: Mathematical notations used in the paper.

## 3 PROBLEM DESCRIPTION

Let $\mathbf{x} \in \mathbb{R}^d$ represent the $d$-dimensional input feature vector [1], and $y \in \{1, 2, ..., C\}$ denote the associated target label, where $C$ is the number of classes. Additionally, let $M \subseteq [d] \equiv \{1, ..., d\}$ be the subset of indices indicating the available features, and $\mathbf{x}_M$ be the masked input vector with these available features. Each feature $j$ has an associated cost $c_j$, and each input $\mathbf{x}$ is subject to a budget constraint $k$. The objective is to find a predictor $f_\theta$, parameterized with $\theta$, and a policy network $q_\pi$, parameterized with $\pi$, such that the following constraint objective is minimized:

$$\min_{\theta, \pi} \mathbb{E}_{\mathbf{x}yk} \mathbb{E}_{M \sim q_\pi} [\ell(f_\theta(\mathbf{x}_M), y)], \text{ s.t.} \sum_{j \in M} c_j \leq k, \tag{1}$$

where the first expectation is taken over the joint distribution of $\mathbf{x}$, $y$, and $k$. The subset $M$ is generated sequentially by the policy network $q_\pi$, which determines the next missing feature to acquire, i.e., $\arg\max q_\pi(\mathbf{x}_M) \in [d] \backslash M$. And, the predictor $f_\theta$ makes probabilistic predictions for any possible subset $M$, i.e., $f_\theta(\mathbf{x}_M) \in [0, 1]^{C,1}$. For brevity, let the output of $q_\pi$ be denoted as $\hat{\mathbf{q}}$, i.e., $\hat{\mathbf{q}} = q_\pi(\mathbf{x}_M)$ and the output of $f_\theta$ be denoted as $\hat{\mathbf{y}}$, i.e., $\hat{\mathbf{y}} = f_\theta(\mathbf{x}_M)$.

---

[1]Each feature can have different dimension size but ease of exposition, in here we have assumed each feature is one dimensional.

Typically, methods in the literature (Yin et al., 2020; Covert et al., 2023b) assume that features have identical costs and that there is a fixed global budget $k$ for all inputs. Given the available training samples $\{(\mathbf{x}^i, y^i)\}_{i=1}^N$, these methods aim identifying input-specific important features to acquire them sequentially in order of the most informative feature to the least one. To achieve this, they train $q_\pi$ through exploration using reinforcement learning (RL) (Yin et al., 2020) or information-theoretic (Covert et al., 2023b) formulations, while simultaneously training the predictor network $f_\theta$.

In this paper, we approach the problem from a different perspective by assuming having access to feature importance rankings for each training sample. Consequently, instead of treating it as a feature exploration problem, we address it as a feature prediction problem (Figure 1).

## 4 OUR METHODOLOGY

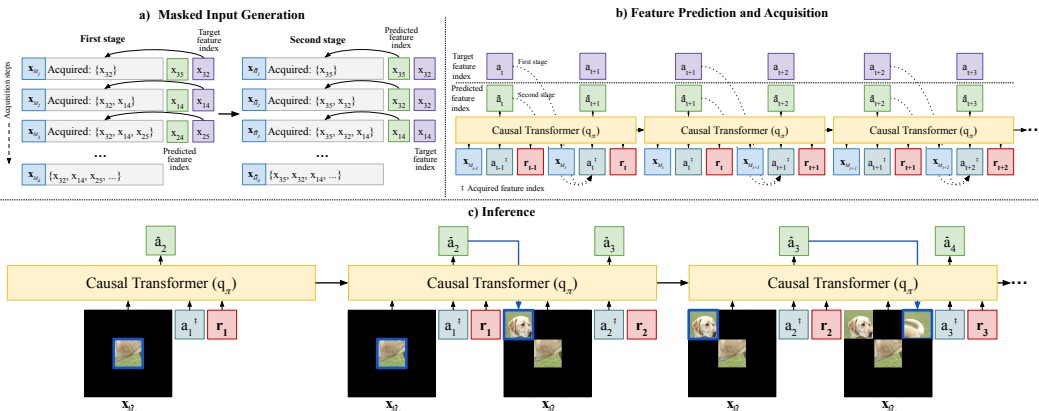

Figure 1: **Overview of our active feature acquisition framework.** a) Our training strategy consists of two stages and this figure shows how the masked inputs are generated during the first and second stages. In the first stage, features are selected based on their ranking order derived from SHAP values. In the second stage, features are acquired by the policy network ($q_\pi$). During the first stage, the next feature in the ranking is the target feature index. However, in the second stage, the target feature is the feature index having the highest SHAP value among the ones that are not acquired; because of this, the target feature remains the same until it is acquired. b) This part of the figure shows how the policy network $q_\pi$, based on the decision transformer (Chen et al., 2021), processes the masked inputs during training. Sequential data with a context length $\ell$, set to 2 in this case, is fed into $q_\pi$. At each time step, $q_\pi$ receives three tokens: the masked input ($\mathbf{x}_{M_t}$), action ($a_t^{(i)}$) and reward ($\mathbf{r_t}$). The action token represents the index of the last acquired feature, and the reward is the output of the predictor network. To ensure causality, future tokens are masked while $q_\pi$ predicts the next feature to acquire at any time step. c) This figure illustrates the inference stage for image inputs in the causal transformer model, where predicted features (or patches) are progressively acquired in a series of sequential acquisition steps.

**Feature importance ranking**. In our method, we assume access to the feature rankings $\varphi^i$ for each training sample $\mathbf{x}^i$, sorted by their importance. While determining the importance of features for each input is challenging, we found that local explanation methods, particularly SHAP (Lundberg & Lee, 2017), can effectively achieve this goal. ~~Empirically, we observed~~ We assumed that a model with reasonable task performance would naturally prioritize the most important instance-specific features, which can be identified using explanation methods. We empirically validated our assumption by observing that if the policy network perfectly acquires features in the order of highest to lowest absolute SHAP values sequentially during inference, the predictor achieves the best performance on average for a given budget of $k$ available features, compared to the current state of the art methods (Figure 2).

To be able to get the SHAP values of the features for the each input, first, we train a classifier using $\{(\mathbf{x}^i, y^i)\}_{i=1}^N$ with the standard cross-entropy loss minimization. Then, we calculate the SHAP values

| Dataset | # Features ($d$) | # Classes | # Samples | Image size | Patch size |
|---|---|---|---|---|---|
| ImageNette | 196 | 10 | 13,395 | $224 \times 224$ | $16 \times 16$ |
| CIFAR-100 | 64 | 100 | 60,000 | $32 \times 32$ | $4 \times 4$ |
| CIFAR-10 | 64 | 10 | 60,000 | $32 \times 32$ | $4 \times 4$ |
| BloodMNIST | 196 | 8 | 17,092 | $28 \times 28$ | $2 \times 2$ |
| Spambase | 57 | 2 | 4,601 | - | - |
| Metabric | 489 | 6 | 1,898 | - | - |
| CKD | 50 | 2 | 1,659 | - | - |
| CPS | 8 | 3 | 418 | - | - |
| CTGS | 23 | 2 | 2,139 | - | - |

Table 2: **Summary of datasets used in our experiments.** For each dataset, we listed the number of features ($d$), number of classes, number of samples, image size, and patch size utilized. Note that Spambase is a tabular dataset; therefore, image size and patch size are not applicable.

of the features for each input $\mathbf{x}^i$ and sort them to get $\varphi^i$, where $\varphi^i(1)$ is the feature index having the highest absolute SHAP value and $\varphi^i(d)$ is the feature index having the lowest absolute SHAP value for the input $\mathbf{x}^i$. So our training set is $\{(\mathbf{x}^i, y^i, \varphi^i)\}_{i=1}^N$.

**Policy network - Decision transformer**. By approaching the problem as the conditional sequence modeling task, similar to in the "decision transformer" framework (Chen et al., 2021), we train $q_\pi$, which is a causal transformer model, with the objective of next action/token prediction. We feed $q_\pi$ with sequential data with a sequence length (i.e., context length) of $\ell$. At each timestep, there are three tokens including the input, the action and the reward as described in Chen et al. (2021). During training, at the timestep of $t$, the input is $\mathbf{x}_{M_t}^i$, which is the $i$'th sample with the $t$ many available features and $M_t = \{\varphi^i(1), ..., \varphi^i(t)\}$[2]. Whereas, the action $a_t^i$ is the most recently acquired feature index, i.e., $a_t^i = \varphi^i(t)$ and the reward $\mathbf{r}_t^i$ is the output of the predictor with the current input, i.e., $\mathbf{r}_t^i = \hat{\mathbf{y}}_t^i = f_\theta(\mathbf{x}_{M_t}^i)$. The rewards in reinforcement learning-based methods (Kachuee et al., 2019; Li & Oliva, 2021) are typically functions of the predictor output; in our method, we follow a similar idea, but instead of defining a specific function, we directly feed our policy transformer network with the predictor output. So, for a given sequence from the timestep $t$ to $t + \ell - 1$, the output of our $q_\pi$ for the input $i$ is: $\hat{\mathbf{q}}_t^i = q_\pi(\mathbf{x}_{M_t}^i, a_t^i, \mathbf{r}_t^i)$ and $\hat{\mathbf{q}}_{t+\ell-1}^i = q_\pi(\mathbf{x}_{M_{t:t+\ell-1}}^i, a_{t:t+\ell-1}^i, \mathbf{r}_{t:t+\ell-1}^i)$, where $t : t + \ell - 1$ indicates all the tokens from the time step $t$ to $t + \ell - 1$. We used a mini version of GPT[3] architecture (Radford, 2018) as a transformer model. Please refer to the decision transformer paper (Chen et al., 2021) for more details.

**Training strategy**. To train $q_\pi$, we minimized the standard cross-entropy loss by considering the index of the next feature that is not acquired with the highest SHAP value (i.e., $\varphi^i(t + 1)$) as the true label with the minibatch setting. At each iteration, the loss function is:

$$\mathcal{L}_q = -\frac{1}{N_b} \sum_{i=1}^{N_b} \sum_{t=t_i}^{t_i+\ell-1} \log(\hat{\mathbf{q}}_{t,\varphi^i(t+1)}^i), \tag{2}$$

where $N_b$ is the batch size, $\hat{\mathbf{q}}_{t,\varphi^i(t+1)}^i$ is the $\varphi^i(t + 1)$'th element of $\hat{\mathbf{q}}_t^i$, and $t_i$ is randomly sampled integer determining the initial time step of sequence fed to the model for the $i$'th sample. Simultaneously, we train the predictor $f_\theta$ also by minimizing the standard cross-entropy loss:

$$\mathcal{L}_f = -\frac{1}{N_b} \sum_{i=1}^{N} \sum_{t=t_i}^{t_i+\ell-1} \log(\hat{\mathbf{y}}_{t,y}^i). \tag{3}$$

During the first stage of training, both $f_\theta$ and $q_\pi$ are fed by the input with the features that are acquired based on the SHAP value ranking order. However during inference, because $q_\pi$ is not 100% accurate, the feature subset $\hat{M}_t$, generated by $q_\pi$, may not always contain the top $t$ features with the

---

[2]Each sample $i$ has its own specific $M_t$, but we do not specify through superscript $i$ if it is clear from the context.

[3]https://github.com/karpathy/minGPT

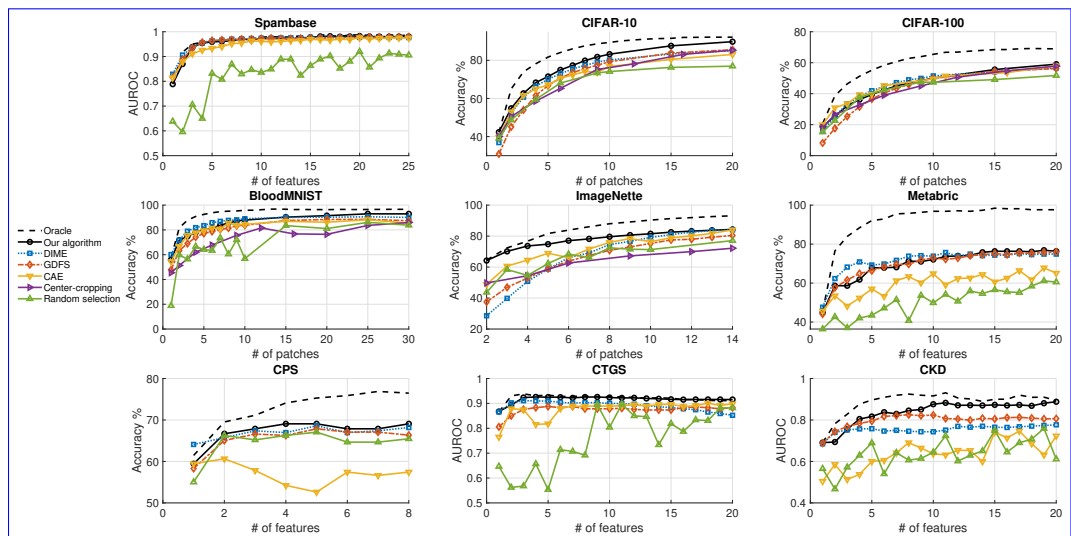

Figure 2: **Model performance.** Average classification performance of our AFA method compared with other well-known methods across varying number of features on the Spambase tabular and four image datasets: CIFAR-10, CIFAR-100, BloodMNIST and ImageNette.

highest SHAP values. To train both models to handle this new subset of features not encountered in the first stage, we introduce a second stage of training. At the beginning of each iteration of the second stage, we first generate empiric/predicted feature acquisition $\hat{\varphi}^i$ order for each $\mathbf{x}^i$, where $\hat{\varphi}^i(t+1) = \arg\max \hat{\mathbf{q}}_t^i$ and $\hat{M}_t = \{\hat{\varphi}^i(1), \hat{\varphi}^i(2), ..., \hat{\varphi}^i(t)\}$. Then, we minimize the same losses as in the first stage with the same strategy. In $\mathcal{L}_q$, the index of the feature, which is not acquired yet and having the highest SHAP value among the features that are not acquired, is taken as the true label. For example, if $\varphi^i(1) \notin \{\hat{\varphi}^i(1), ..., \hat{\varphi}^i(t)\}$ then $\varphi^i(1)$ is taken as the true label; but if $\varphi^i(1)$ is acquired and $\varphi^i(2)$ is not acquired then $\varphi^i(2)$ is taken as the true label, i.e., $\varphi^i(1) \in \{\hat{\varphi}^i(1), ..., \hat{\varphi}^i(t)\}$ and $\varphi^i(2) \notin \{\hat{\varphi}^i(1), ..., \hat{\varphi}^i(t)\}$. By this second stage, we train the predictor $f_\theta$ to make its prediction with the subset of features $\hat{M}_t$ acquired by $q_\pi$. Also, the policy network $q_\pi$ is trained to predict the feature with the highest SHAP value among the features that are not acquired using the input with the imperfect subset of features $\hat{M}_t$. This second stage helps both networks to perform better during inference, where the imperfect subset of features $\hat{M}_t$ can only be used. Note that both the predictor and policy networks are dependent on each other. However, during training, we prevent the gradient flow from one network to another. Therefore, each network has its own independent loss function; because of the dependency, we trained them simultaneously. At $t = 0$, there is no feature acquired yet, i.e., $M_0 = \emptyset$; so for all $i$, the outputs of $q_\pi$ are the same at $t = 0$. Consequently, at $t = 0$, for all inputs we have to choose the same feature to be acquired. In our method, we initialized each input by the fixed first feature that has the highest SHAP value on average calculated on the training set.

**Implementation details**. During training, we set number of epochs to 200 and 16 for the first and second stage, respectively. We used Adam optimizer (Kingma & Ba, 2014) and a cosine scheduler (Loshchilov & Hutter, 2017). Before starting training, we pre-trained the predictor network, as done in (Covert et al., 2023b; Gadgil et al., 2024). We also employed a different augmentation strategy proposed in (Hoffer et al., 2020). Also, as done by other methods in the literature (Kachuee et al., 2019; Covert et al., 2023b; Gadgil et al., 2024), we shared the backbone between $f_\theta$ and $q_\pi$. We used this backbone in $q_\pi$ to get the embedding of the input token. The embedding of action was extracted using a learnable embedding dictionary. For the reward's embedding, a simple MLP was used. In $q_\pi$, we set context length $\ell$ to 4, number of heads and layers 4 and 3, respectively.

| | Spambase | CIFAR-10 | CIFAR-100 | BloodMNIST | ImageNette |
|---|---|---|---|---|---|
| # of classes: | 2 | 10 | 100 | 8 | 10 |
| First-stage | 0.9512 | 75.68% | 45.88% | 79.08% | 73.35% |
| Second-stage | 0.9559 | 78.61% | 47.06% | 84.38% | 79.08% |

Table 3: **Stage-wise classification performance.** The table presents our model's performance after the first and second training stages, averaged over the first 20 features, on the Spambase, CIFAR-10, CIFAR-100, BloodMNIST, and ImageNette datasets. For the Spambase dataset, we reported the area under the receiver operating characteristic curve values, while for the remaining datasets, we provided accuracy metrics.

| | CIFAR-10 | CIFAR-100 | BloodMNIST | ImageNette |
|---|---|---|---|---|
| # of features ($d$): | 64 | 64 | 196 | 196 |
| Top 10 features | 36.54% | 48.71% | 42.17% | 11.08% |
| Top 15 features | 46.08% | 58.30% | 49.09% | 15.99% |
| Top 20 features | 52.46% | 64.78% | 53.58% | 20.61% |
| Top 25 features | 57.41% | 68.71% | 56.52% | 24.92% |
| Top 30 features | 62.00% | 71.32% | 58.63% | 29.01% |

Table 4: **Alignment between model's feature acquisition order and the SHAP-based feature importance rankings.** This table presents the percentage overlap between the top N features ranked by SHAP values and features acquired by our model for N = 10, 15, 20, 25, and 30. The datasets include CIFAR-10 and CIFAR-100 (each with 64 features), and BloodMNIST and ImageNette (each with 196 features).

## 5 RESULTS AND DISCUSSION

We utilized several datasets in our experiments (Table 2), including ImageNette, CIFAR-10, CIFAR-100, BloodMNIST, and Spambase. ImageNette ~~Howard (2019)~~ (Howard, 2019) is a 10-class subset of the ImageNet dataset ~~Deng et al. (2009)~~(Deng et al., 2009). CIFAR-10 and CIFAR-100 ~~Krizhevsky (2009)~~ (Krizhevsky, 2009) are subsets of the 80 Million Tiny Images dataset ~~Torralba et al. (2008)~~(Torralba et al., 2008), containing 10 and 100 classes respectively. BloodM-NIST (Acevedo et al., 2020), derived from the MedMNIST dataset (Yang et al., 2021; 2023), comprises images of individual normal cells collected from individuals without infection, hematologic or oncologic diseases, and free of any pharmacologic treatment at the time of blood collection. The patch sizes are $16 \times 16$ for ImageNette (makes total of 196 patches, $d = 196$), $4 \times 4$ for the CIFAR-10 and CIFAR-100 datasets ($d = 64$), and $2 \times 2$ for the BloodMNIST dataset ($d = 196$). Spambase (Hopkins & Suermondt, 1999) is a well-known tabular dataset for classifying spam emails, consisting of 57 features derived from textual data. Additionally, to assess the applicability of our method in real-world scenarios, such as healthcare, we conducted experiments on four medical tabular datasets. As part of the preprocessing, we removed ID columns and categorical columns that were not ranking-based or binary. Columns with more than 10% missing values were also excluded, while the remaining missing values were imputed with the mean. In the following and in Table 2, the number of features refers to the count after preprocessing. The Metabric dataset (Curtis et al., 2012; Pereira et al., 2016) contains targeted gene sequencing data from 1,898 breast cancer samples, where we utilized mRNA-level Z-scores, which contains 489 features, to predict the Pam50 gene status that is a multi-class classification task. The Cirrhosis Patient Survival (CPS) dataset (Dickson & Langworthy, 1989) includes records from 418 patients, primarily with primary biliary cirrhosis, along with 8 clinical features, with the task of predicting patient survival states categorized as Death, Censored, or Censored Due to Liver Transplantation. The AIDS Clinical Trials Group Study 175 (CTGS) dataset (Hammer et al., 1996) contains 2139 records of patients diagnosed with AIDS, 23 features, with a binary classification task to predict whether a patient has died within a specified time period. Lastly, the Chronic Kidney Disease (CKD) dataset (Kharoua, 2024) comprises 1659 patient records with 50 clinical features, and the task is to predict whether a patient is diagnosed with chronic kidney disease in a binary classification setting.

| # of classes: | **Metabric** 6 | **CPS** 3 | **CTGS** 2 | **CKD** 2 |
|---|---|---|---|---|
| TreeSHAP | 69.90% | 67.12% | 0.9167 | 0.8473 |
| LIME | 69.04% | 66.21% | 0.9125 | 0.8122 |
| KernelSHAP | 70.01% | 66.06% | 0.9146 | 0.8353 |
| Sampling(IME) | 69.72% | 65.91% | 0.9160 | 0.8152 |

Table 5: **Model performance using various feature ranking approaches.** Comparison of classification performance across four medical datasets using feature rankings derived from various local explanation methods: TreeSHAP, LIME, KernelSHAP, and Sampling (IME). The performance metrics are the area under the receiver operating characteristic curve for the binary-classification datasets and accuracy for the multi-class datasets.

To test the robustness of our method across different architectures, we also varied predictor architectures. We employed ResNet50 ~~He et al. (2016)~~ (He et al., 2016) for ImageNette, ResNet18 ~~He et al. (2016)~~ (He et al., 2016) for the CIFAR-10 and CIFAR-100 datasets, and a custom CNN for the BloodMNIST dataset. The custom CNN architecture has four convolution layers with output channels $16, 32, 64,$ and $64$, each followed by a ReLU activation and a max pooling layer. The convolution layers are followed by flattening and linear layers for classification. For the Spambase dataset, we used a multi-layer perception (MLP) architecture consisting of 2 hidden layers with 128 neurons, each followed by a ReLU and a dropout layer. On the medical tabular datasets, we utilized the same MLP architecture with 1024 hidden layer neurons on Metabric, 512 on CKD, 512 on CTGS and 128 on CPS. For the image datasets, we employed FastSHAP (Jethani et al., 2022) to generate the feature SHAP ranking order $\varphi^i$ for each instance $\mathbf{x}^i$ due to its speed. During training, we applied random augmentations that can affect feature importance. The speed of FastSHAP allows us to efficiently handle these changes in feature importance during the training process. For the ~~Spambase dataset~~ tabular datasets, we did not apply any data augmentation during training. ~~We obtained the~~ Tree-based models, specifically CatBoost (Prokhorenkova et al., 2018), were used as the initial model to determine feature ranking orders, owing to their superior performance on tabular data (Grinsztajn et al., 2022). SHAP ranking orders were obtained using the SHAP package[4] ~~.~~ , leveraging TreeSHAP (Lundberg et al., 2020), a method specifically designed for SHAP value calculations in tree-based models

We evaluated our method against several existing approaches for feature selection: Discriminative Mutual Information Estimation (DIME), Greedy Dynamic Feature Selection (GDFS), Concrete Autoencoder (CAE), and two simple baselines ~~—~~: center-cropping and random selection. DIME (Gadgil et al., 2024) takes an information-theoretic approach by prioritizing features based on their mutual information with the response variable, estimating this mutual information in a discriminative rather than a generative manner. GDFS (Covert et al., 2023b) employs a simpler, greedy strategy for selecting features based on their conditional mutual information, utilizing a learning approach grounded in amortized optimization; the policy network is shown to recover the greedy policy when trained to optimality. CAE (Balın et al., 2019) is an unsupervised, end-to-end differentiable method for global feature selection that uses a standard neural network as the decoder for reconstruction and incorporates a concrete selector layer as the encoder. The temperature in CAE is gradually decreased to slowly discretize the selections. The center-cropping and random selection methods (Covert et al., 2023b) serve as simple baselines: center-cropping selects center patches of varying sizes, while random selection chooses patches randomly.

Figure 2 demonstrates that our method shows superior, or comparable performance on all the datasets. For example, on the ImageNette dataset, with the few number of patches, our method performs well, achieving $64.2\%$ and $74.8\%$ average accuracy with two and five available patches among 196 patches, respectively. Additionally, our model achieved an AUROC score of 0.8761 on the CKD dataset. To demonstrate the relative potential of our approach, we also provided the oracle setting performances, where during inference the features are perfectly acquired based on the SHAP ranking. On the oracle setting, we also initialized the instances with the feature having the highest SHAP value on average, as in our method. Therefore, while it is theoretically possible to achieve these performance

---

[4]https://pypi.org/project/shap/

levels, empirically we could not attain them with our current method. We discovered that instead of initializing the inputs with only one feature, it is more effective for stable training to begin with three features. Based on this finding, we fixed first three feature acquisition order and we obtained the results as shown in Figure 2. For the initial features, we selected the second and third features based on their average importance. Note that fixing the acquisition order for all $d$ features is equivalent to using static global feature selection methods like CAE, which is suboptimal, as our empirical results demonstrate. Therefore, initializing with more than one feature can negatively impact the achievable upper bound in performance. However, we found that fixing the acquisition order for a few initial features helps stabilize training. Additionally, since our method relies on the feature ranking order, having a better ranking can lead to improved performance. Our approach can work with any ranking order, including those provided by humans, but we have shown that local model explanation algorithms, particularly SHAP, are effective in providing this order.

The average performance after both stages is shown for all the datasets (Table 3), highlighting the benefit of the second stage. The second stage provides significant improvement on almost all datasets, except on the Spambase that is a relatively simpler dataset compared to others, at least in terms of number of classes and features. Specifically, the second stage provides classification accuracy increase from $2.57\%$ (on CIFAR-100) to $7.81\%$ (on ImageNette).

~~Lastly, in Table~~ We also performed some ablation experiments to evaluate the robustness and effectiveness of our method. Firstly, we tested our method with alternative feature ranking approaches, including another explainability method, LIME (Ribeiro et al., 2016), and two different SHAP value calculation techniques: KernelSHAP (Lundberg & Lee, 2017) and IME (sampling) (Štrumbelj & Kononenko, 2010). These results (Table 5) indicate that while our method is robust to different ranking orders, its performance is also dependent on the the quality of the ranking order generated by the explainability methods. To further verify the second point and test the dependency of the SHAP ranking orders' quality on the pre-trained model capacity, we conducted another ablation experiment on the CIFAR-10 dataset. Specifically, we used ResNet-10, a smaller model compared to ResNet-18, as the pre-trained model for determining the SHAP ranking order, while retaining ResNet-18 as the classification network. We observed that the performance of our method decreased from $78.61\%$ to $78.22\%$ on the test set, and from $79.12\%$ to $78.42\%$ on the validation set. These results confirm that the pre-trained model's capacity impacts the SHAP-based ranking order and, consequently, the performance of our method. In addition, we evaluated the effectiveness of using the decision transformer by comparing our method's performance with different architectures. When the decision transformer was replaced with a ResNet block, the model's accuracy decreased from $78.61\%$ to $76.83\%$ on the CIFAR-10 dataset and from $47.06\%$ to $46.70\%$ on the CIFAR-100 dataset. Similarly, substituting the decision transformer with a CNN block reduced the model's accuracy from $84.38\%$ to $78.23\%$ on the BloodMNIST dataset. These results demonstrate the advantage of using a decision transformer as the policy network while also highlighting that our method performs reasonably well with other architectures as the policy network.

~~(Lastly, i)~~In Table 4, we present the overlap ratios between our model's acquired feature order and the SHAP-based feature importance rankings across different datasets. As the number of top features (N) increases from 10 to 30, the percentage overlap generally rises for CIFAR-10, CIFAR-100, BloodMNIST, and ImageNette. This trend indicates that our model's feature acquisition order increasingly aligns with the SHAP rankings as more features are considered. While the oracle performances in Figure 2 demonstrate the practical benefits of using SHAP values in the AFA problem, Table 4 highlights the degree to which our model's acquisition strategy predicts the SHAP-based feature importance ranking. Additionally, we would like to note that we did not perform detailed parameter search on the experiments. We selected the context length $\ell$ parameter by comparing the validation scores on the CIFAR-10 dataset and the other parameters were selected heuristically by hand. Subsequently, all these parameters were fixed for all the experiments. About the $\ell$ parameter selection, we found that our model achieved accuracies of $78.41\%$, $78.76\%$, $79.12\%$, and $78.12\%$ on the CIFAR-10 validation dataset with $\ell = 1$, $\ell = 2$, $\ell = 4$, and $\ell = 8$, respectively. When we increased or decreased the context length $\ell$, we correspondingly varied the batch size $N_b$ by the same factor to maintain the same effective size at each iteration (see Equations 2 and 3). Based on these observations, we assigned $\ell = 4$. Finally, we would like to emphasize that our proposed method is flexible and can operate with any given feature order. However, due to the absence of a reference standard (or ground truth) for feature importance rankings, we relied on explainability methods to generate the feature orders. While these approaches are useful, they

may not always provide good rankings in all scenarios (Kumar et al., 2020; Catav et al., 2021). As different local explanation methods that provide better rankings are developed, our approach can be readily integrated to deliver enhanced results.

Top 10 features 36.54% 48.71% 42.17% 11.08% Top 15 features 46.08% 58.30% 49.09% 15.99% Top 20 features 52.46% 64.78% 53.58% 20.61% Top 25 features 57.41% 68.71% 56.52% 24.92% Top 30 features 62.00% 71.32% 58.63% 29.01% **Alignment between model's feature acquisition order and the SHAP-based feature importance rankings.** This table presents the percentage overlap between the top N features ranked by SHAP values and features acquired by our model for N = 10, 15, 20, 25, and 30. The datasets include CIFAR-10 and CIFAR-100 (each with 64 features), and BloodMNIST and ImageNette (each with 196 features).

## 6 CONCLUSION

Our work ~~proposes a novel~~ introduces an explainability-based active feature acquisition strategy by reframing it as a feature prediction task, where the model learns to acquire features based on instance-specific SHAP value rankings. Stage-wise results ~~show~~ demonstrate that our two stage training approach ~~enhances~~ improves feature selection and classification performance on ~~both~~ tabular and image datasets. The findings ~~also indicate~~ further suggest that our method is robust across ~~varying~~ various models, datasets and settings. ~~Future work could apply our method to more practical datasets, such as those in medical diagnosis, to evaluate its usability in~~ Additionally, our experimental results on medical tabular datasets highlight the practical applicability of our method in real-world scenarios ~~.~~ like healthcare. Future work could explore dynamic recalculations of feature attributions during training after each feature acquisition step, replacing the current use of fixed ordering.

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

## A  APPENDIX

Below we provide pseudocodes for our first and second training stages.

---

**Algorithm 1** Pseudocode for first stage training of $q_\pi$ and $f_\theta$

---

**Require:** Training set $\{(\mathbf{x}^i, y^i, \varphi^i)\}_{i=1}^N$, batch size $N_b$, context length $\ell$, learning rate $\gamma$

1: Pre-train $f_\theta$ on $\{(\mathbf{x}^i, y^i)\}_{i=1}^N$ using random feature selection
2: Initialize $q_\pi$
3: **for** each epoch **do**
4:     **for** 1 to $\lceil N/N_b \rceil$ **do**
5:         Sample minibatch $\{(\mathbf{x}^i, y^i, \varphi^i)\}_{i=1}^{N_b}$ (if random augmentation is applied, $\varphi^i$ is recalculated for each iteration after sampling)
6:         Sample random integer $t_i$ for each $i$
7:         Initialize $\mathcal{L}_q = 0$ and $\mathcal{L}_f = 0$
8:         **for** $t_x = 0$ to $\ell - 1$ **do**
9:             Define temporary parameter $t_i'$ for each $i$, $t_i' = t_i + t_x$
10:             Generate masked input $\mathbf{x}_{M_{t_i'}}^i$, $M_{t_i'} = \{\varphi^i(1), \ldots, \varphi^i(t_i')\}$
11:             Compute predictor output: $\hat{\mathbf{y}}_{t_i'}^i = f_\theta(\mathbf{x}_{M_{t_i'}}^i)$
12:             Compute policy network output: $\hat{\mathbf{q}}_{t_i'}^i = q_\pi(\mathbf{x}_{M_{t:t_i'}}^i, a_{t:t_i'}^i, \mathbf{r}_{t:t_i'}^i)$
13:             Update losses: $\mathcal{L}_f \leftarrow -\frac{1}{N_b}\sum_{i=1}^{N_b} \log(\hat{\mathbf{y}}_{t_i', y^i}^i) + \mathcal{L}_f$
14:             $\mathcal{L}_q \leftarrow -\frac{1}{N_b}\sum_{i=1}^{N_b} \log(\hat{\mathbf{q}}_{t_i', \varphi^i(t_i'+1)}^i) + \mathcal{L}_q$
15:         Update parameters $\theta \leftarrow \theta - \gamma\nabla_\theta\mathcal{L}_f$, $\pi \leftarrow \pi - \gamma\nabla_\pi\mathcal{L}_q$

---

**Algorithm 2** Pseudocode for second stage training of $q_\pi$ and $f_\theta$

---

**Require:** Training set $\{(\mathbf{x}^i, y^i, \varphi^i)\}_{i=1}^N$, batch size $N_b$, context length $\ell$, learning rate $\gamma$, $f_\theta$ and $q_\pi$ from the first stage

1: **for** each epoch **do**
2:     **for** 1 to $\lceil N/N_b \rceil$ **do**
3:         Sample minibatch $\{(\mathbf{x}^i, y^i, \varphi^i)\}_{i=1}^{N_b}$ (if random augmentation is applied, $\varphi^i$ is recalculated for each iteration after sampling)
4:         Generate $\hat{\varphi}^i$ for each $i$
5:         Sample random integer $t_i$ for each $i$
6:         Initialize $\mathcal{L}_q = 0$ and $\mathcal{L}_f = 0$
7:         **for** $t_x = 0$ to $\ell - 1$ **do**
8:             Define temporary parameter $t_i'$ for each $i$, $t_i' = t_i + t_x$
9:             Generate masked input $\mathbf{x}_{\hat{M}_{t_i'}}^i$, $\hat{M}_{t_i'} = \{\hat{\varphi}^i(1), \ldots, \hat{\varphi}^i(t_i')\}$
10:             Compute predictor output: $\hat{\mathbf{y}}_{t_i'}^i = f_\theta(\mathbf{x}_{\hat{M}_{t_i'}}^i)$
11:             Compute policy network output: $\hat{\mathbf{q}}_{t_i'}^i = q_\pi(\mathbf{x}_{\hat{M}_{t:t_i'}}^i, a_{t:t_i'}^i, \mathbf{r}_{t:t_i'}^i)$
12:             Update losses: $\mathcal{L}_f \leftarrow -\frac{1}{N_b}\sum_{i=1}^{N_b} \log(\hat{\mathbf{y}}_{t_i', y^i}^i) + \mathcal{L}_f$
13:             Determine the true label for the $q_\pi$ network (denote this true label as $y_{q_{t_i'}}^i$). The true label is the index of the feature, which is not acquired yet and having the highest SHAP value among the features that are not acquired
14:             $\mathcal{L}_q \leftarrow -\frac{1}{N_b}\sum_{i=1}^{N_b} \log(\hat{\mathbf{q}}_{t_i', y_{q_{t_i'}}^i}^i) + \mathcal{L}_q$
15:         Update parameters $\theta \leftarrow \theta - \gamma\nabla_\theta\mathcal{L}_f$, $\pi \leftarrow \pi - \gamma\nabla_\pi\mathcal{L}_q$

---

