# OpenReview forum: "Interpretability-driven active feature acquisition in learning systems"
_ICLR.cc/2025/Conference — ICLR 2025 Conference Withdrawn Submission_

### Official Review · Reviewer_9rQA · 2024-10-31

**Soundness:** 3
**Presentation:** 3
**Contribution:** 3
**Rating:** 3
**Confidence:** 4

**Summary:**

This paper introduces a new approach to active feature acquisition (AFA) that uses SHAP to determine which feature to acquire next given instance. The key idea is to use pre-specified SHAP values on the training set obtained using a pretrained classifier, and then use these values to train a policy network to predict the next most informative feature based on these SHAP values. The authors show that their approach outperforms current state-of-the-art AFA techniques on multiple datasets (4 image datasets and 1 tabular dataset).

**Strengths:**

-	The paper is in general well-written and easy to follow.
-	The authors provide extensive quantitative analysis by thoroughly comparing the proposed method to a wide range of feature selection methods.
-	The two-stage training approach and the use of SHAP values to guide feature acquisition sounds like a promising direction.
-	The proposed method provides state-of-the-art performance.

**Weaknesses:**

-	The authors don't fully justify why they use SHAP values to guide feature acquisition. Although their "Oracle" method, which uses the ground-truth SHAP, seems effective, SHAP might not be the best way to choose the most informative features. SHAP measures how much each feature contributes to a change in the model's prediction, but this doesn't necessarily mean those features are the best for making accurate predictions with only a few features.
-	The approach might not fully consider how the importance of acquiring a feature can change based on what features are already acquired. The ordering of SHAP value is fixed during training and thus the policy network (even though it takes the sequence of observed features as input) will neglect such dependencies, which could be the key to AFA.
-	The authors motivated the impact of AFA on healthcare and medical applications, but only one out of five datasets is healthcare-related. The other datasets are general image datasets where the artificial partitioning into smaller patches doesn't reflect real-world scenarios. Using real-world tabular medical datasets (like MIMIC or METABRIC) would better demonstrate the method's relevance to its stated application.
-	The authors do not clearly justify why a decision transformer is used as the policy network and the entire sequence of reward is used as input. It's unclear if performance gains are from this specific architecture or from the SHAP guidance.
-	More in-depth case studies would strengthen the paper. Showing how and why feature acquisition order varies across different samples would provide valuable insights.
-	Training Strategy: The details of the training process, especially how the context length ($\ell$) is chosen and how pre-selection is used, could be clearer. Pseudo-code would improve understanding and reproducibility.

**Questions:**

-	Regarding weakness 1, please elaborate the underlying motivation of using the ordering of SHAP value for training the policy network. For instance, why the features that contribute to the model prediction the most should be most discriminative and thus suitable for AFA?
-	The SHAP ordering of features is treated as a fixed ground truth. However, in reality, the order in which features are most informative might shift depending on the information already acquired. For example, if the model already has access to a highly predictive feature, the relative importance of other features might decrease or change. The current approach, by relying on a pre-specified SHAP ordering, could overlook these dynamic interactions between features, potentially leading to suboptimal feature acquisition choices. Could the authors explain why this is not an issue?
-	How does the policy network prevent the selection of an already-chosen feature? I couldn’t identify any explicit architecture design or loss to prevent the re-selection of previously acquired features as the policy network simply provides a softmax output at each instance.
-	To better illustrate the advantages of the proposed SHAP-based AFA method, could the authors provide some case studies? It would be helpful to see examples where traditional AFA methods fail, but the proposed method, guided by SHAP ordering, produces more accurate (discriminative) predictions. Additionally, consider including a subgroup analysis based on similarities in feature acquisition order. This would help to demonstrate the effectiveness of using SHAP values in guiding AFA.

---

> ### Author Response · Authors · 2024-11-26
> **Response to 9rQA's comments**
>
> Response to W1 & Q1: We thank the reviewer for this insightful comment. As noted, our conclusions are primarily based on empirical observations derived from the Oracle results, which demonstrate that our approach outperforms other AFA methods. Intuitively, we assumed that a model with reasonable task performance would inherently emphasize the most important features, and this hypothesis was empirically validated through our Oracle experiments. However, we acknowledge the limitations of SHAP as a feature ranking method, including its dependency on the underlying model and the concerns highlighted by other reviewers. To further validate our assumptions, we conducted an ablation experiment on the CIFAR-10 dataset. Specifically, we used ResNet-10, a smaller model compared to ResNet-18, as the pre-trained model for determining the SHAP ranking order, while retaining ResNet-18 as the classification network. We observed that the performance of our method decreased from 78.61% to 78.22% on the test set, and from 79.12% to 78.42% on the validation set. These results indicate that the model’s capacity to emphasize important features impacts the SHAP-based ranking order and, consequently, the performance of our method. This supports our hypothesis that models with strong task performance inherently prioritize important features, even when relying on SHAP-based rankings.
>
> Response to W2: Recalculating feature attribution values dynamically as features are acquired could capture changing dependencies and potentially improve the feature acquisition order. However, recalculating SHAP values dynamically is computationally prohibitive due to the exponential number of subsets involved. Methods such as Kernel-SHAP and FastSHAP approximate SHAP values by sampling a subset of these differences or using deep learning models for efficient estimation. Due to these constraints, we opted to keep the SHAP ordering fixed during training. Despite this limitation, our empirical results demonstrate that the fixed SHAP ordering outperforms state-of-the-art AFA techniques in terms of accuracy for a given number of features (see Oracle performance comparisons in the results section). This suggests that even with a fixed SHAP ordering, our method effectively captures meaningful feature importance rankings for AFA. We acknowledge that exploring dynamic recalculations of feature importance remains an open area for future research, and we appreciate the reviewer for highlighting this potential avenue for improvement. We incorporated this point as a potential direction for future research in the revised manuscript.
>
> Response to W6: We did not perform detailed parameter search on the experiments. We selected the context length $\ell$ parameter by comparing the validation scores on CIFAR-10 and the other parameters were selected heuristically by hand. Subsequently, all these parameters were fixed for all the experiments. About the $\ell$ parameter selection, we found that our model achieved accuracies of 78.41%, 78.76%, 79.12%, and 78.12% on the CIFAR-10 validation dataset with $\ell=1$, $\ell=2$, $\ell=4$, and $\ell=8$, respectively. Based on these observations, we assigned $\ell=4$. We added these details and pseudocodes for our training strategies (in the Appendix) in the revised manuscript.
>
> Response to W3, W4 & W5: Please see the common response.
>
> Response to Q2: The static global baselines often fail compared to the dynamic baselines because static global feature rankings do not inherently capture the interactions between the features at the instance level. This was our motivation behind creating a new AFA framework that acquires features based on instance-specific ranking. While we agree that we can improve our framework by recalculating feature attributions after each acquisition considering the interaction between the features, this approach could be computationally expensive. Our experimental results, especially that of the oracle network, show the effectiveness of how acquiring features based on instance-specific ranking, even without recalculations, can largely improve the performance of the AFA framework. Please check the common response for more information.
>
> Response to Q3: We thank the reviewer for asking this question seeking clarification on how we handle re-acquisition. To select features only from the unacquired feature set, during both training and inference, we subtracted $m \cdot \text{inf}$, where $m$ is the mask and $\text{inf}$, from the output logits of the policy network before applying the softmax layer. This was done to suppress the logits of the acquired features. In the paper, we mentioned that the subset of features within the acquisition pool is taken by a set difference of the features and the mask:
>
> $$\arg\max q_\pi(\mathbf{x}_M) \in [d] \backslash M$$
>
> Due to this, the acquired index $\arg\max q_\pi(\mathbf{x}_M)$ will not be one that has been previously acquired.

---

> > ### Author Response · Authors · 2024-11-28
> > **Response to 9rQA's comments - continued**
> >
> > Response to Q4: While we agree that showcasing the benefits of our method through case studies would add value, we want to clarify that our approach is not inherently tied to SHAP-based feature rankings. By reframing the AFA problem as a feature prediction task, we propose a policy network based on a decision transformer architecture that can work with any feature ranking technique (please see the common response for our method results with using different ranking approaches). While we utilized SHAP values as an example in this work, our framework is designed to adapt to other attribution methods. Also since feature rankings lack a ground-truth (or a reference) standard, we can only compare our method to others relatively, focusing on improvements in downstream task performance and acquisition efficiency.

---

### Official Review · Reviewer_yCGi · 2024-11-03

**Soundness:** 3
**Presentation:** 3
**Contribution:** 2
**Rating:** 6
**Confidence:** 4

**Summary:**

This paper introduces a new approach to active feature acquisition, particularly for scenarios where gathering all features is expensive or time-consuming, such as in medical applications. They introduce the first AFA framework that leverages SHAP values to determine instance-specific feature importance rankings. They reframe AFA as a feature prediction task rather than a feature exploration problem, using a decision transformer architecture to predict which feature to acquire next based on SHAP values. They propose a novel two-stage training approach: First stage: Training using SHAP value ranking order. Second stage: Training using predicted feature acquisition order to improve real-world performance. Through experiments across multiple datasets they demonstrate their method achieves superior or comparable results to state-of-the-art AFA techniques.

**Strengths:**

-The paper reframes AFA from an exploration problem to a prediction problem and leverages SHAP values for determining instance-wise feature importance in AFA, which is novel and innovative

**Weaknesses:**

- The paper lacks theoretical justification for why predicting SHAP values should lead to optimal feature acquisition
-This method bridges between interpretability methods and feature acquisition, it doesn't provide any interpretability results
- No evaluation of robustness to: a) Input perturbations, b) Different model architectures beyond those tested, c) Changes in the SHAP value computation approach

**Questions:**

To improve the paper, I would recommend: Adding detailed analysis of policy network prediction patterns and failure modes, providing thorough ablation studies of the training process

---

> ### Author Response · Authors · 2024-11-26
> **Response to yCGi's comments**
>
> Response to W1: We thank the reviewer for raising this important point. While SHAP may not be the only available method for determining instance-wise feature importance, we empirically observed that an ideal (oracle) policy network that sequentially selects features based on their SHAP values, ranked from highest to lowest, outperforms current state-of-the-art active feature acquisition (AFA) techniques in terms of accuracy for any fixed number of features (see oracle performance comparisons in our results). In the AFA domain, determining the optimal feature acquisition order for each instance remains a significant challenge due to the lack of ground truth (or reference standard) for feature importance rankings. To the best of our knowledge, no prior work provides theoretical guarantees for the optimality of feature acquisition orders. For example, the GDFS method provides theoretical justification for recovering a greedy policy based on its objective functions but does not prove the optimality of the greedy policy itself in the AFA problem. In addition, intuitively, we assumed that a model with reasonable task performance would inherently emphasize the instance-wise most important features, and explanation methods can identify those emphasized features. Through our oracle experiments, we showed that SHAP provides a relatively better ranking order compared to current state-of-the-art methods, making it a practical and effective choice for our approach despite its limitations. These Oracle experiments further support our assumption that models with strong task performance inherently prioritize important features, which can be effectively identified using explanation methods.
>
> Response to W2 & Q1: During training on the image datasets, we applied random augmentations. Because of the superior performance of our method on these datasets, we believe our method is robust to input perturbations. Also, in the experiments, we used three different types of architectures: MLP, CNN, and ResNets (both ResNet18 and ResNet50), which also show that our method can be integrated with various architectures. Additionally, we performed an ablation study to evaluate the effectiveness of using decision transformer and the robustness of our method with different architectures on the policy network. Also, on our new tabular dataset results, we show the performance of our method tested on other feature ranking methods, such as LIME, and different SHAP calculation methods (TreeSHAP vs KernelSHAP). For example, the results with varying approaches of explanation and SHAP calculations show that our model is dependent on the quality of the ranking order provided by the explanation methods.

---

### Official Review · Reviewer_jvZd · 2024-11-03

**Soundness:** 2
**Presentation:** 3
**Contribution:** 2
**Rating:** 3
**Confidence:** 5

**Summary:**

The goal of this paper is to sequentially acquire features, for applications where using all features is costly. To do that, the paper first uses SHAp to generate feature rankings for all training data, and then train a policy network to predict the next most important feature.

**Strengths:**

The problem is interesting and important. The performance of the proposed method is good according to the experiments.

**Weaknesses:**

There are two main issues with the paper.

1. The main motivation for the method is that acquiring all features can be expensive, time-consuming and sometimes have to be done sequentially. This argument convinced me at the introduction when the authors mention medical settings. However, all experiments were done using image data, except one using email spam data. I cannot think of any realistic settings where one can only query one patch at a time from an image, or one feature at a time from emails. The application of these methods to these datasets are not convincing at all. Since the authors mentioned medical setting, why not using medical datasets to prove the value of the method?

2. Another concern is the use of SHAP to determine the feature importance. For this specific setting, what the authors really need to predict is the feature that will has a big impact on the predictive performance of a model, not features that will make biggest contribution to the output. For example, one feature could make biggest contribution, but if it makes almost the same contribution to every instance, then it will have little impact on the predictive performance because knowing that feature does not increase the accuracy much. I think a more appropriate feature importance should be something like mean decrease in accuracy.

**Questions:**

Did you try using other importance metrics like mean decrease in accuracy? Why did you use SHAP?

Can you find more convincing and realistic applications of your method?

---

> ### Author Response · Authors · 2024-11-26
> **Response to jvZd's comments**
>
> Response to W1 & Q2 : We appreciate reviewer's suggestion to explore medical datasets. Please see the common response.
>
> Response to W2 & Q1: We appreciate the reviewer’s thoughtful feedback regarding the use of SHAP and the suggestion to consider metrics like mean decrease in accuracy. Our approach is motivated by the need to balance both global and instance-specific perspectives of feature importance. While mean decrease in accuracy is effective for assessing a feature's global impact on predictive performance, it does not capture instance-level variations, which can be crucial in certain contexts. For example, a feature that is consistently important across all instances might not provide significant additional information for certain model updates, whereas instance-specific variations can highlight features that are critical for refining predictions in localized regions of the feature space. Our acquisition policy leverages an instance-specific ranking of feature importance, which inherently accommodates both scenarios: (a) For features with consistent importance across the dataset, our policy prioritizes them naturally based on their contribution to overall performance. (b) For features with instance-specific importance, our policy effectively adapts to their variability and ensures targeted acquisition that maximizes information gain. To validate this, we implemented an “Oracle” network as a baseline to demonstrate how acquiring features based on instance-specific rankings can substantially enhance the performance of the AFA framework. Our results indicate that instance-specific ranking offers significant benefits over static global approaches. We will provide a detailed discussion of these results in the revised manuscript, emphasizing how our approach balances global and local perspectives while maintaining robustness to the reviewer's highlighted concern.

---

### Official Review · Reviewer_grQh · 2024-11-04

**Soundness:** 2
**Presentation:** 3
**Contribution:** 2
**Rating:** 3
**Confidence:** 4

**Summary:**

This paper proposes an active feature acquisition (AFA) framework leveraging Shapley Additive Explanations (SHAP) values to determine feature importance on an instance-by-instance basis. The authors train a decision transformer model to imitate a SHAP-based oracle, intending to optimize the sequential selection of features to maximize predictive accuracy while minimizing acquisition cost. The model's training follows a teacher-forcing-style imitation learning approach, complemented by an on-policy adaptation stage. Experiments are conducted on various image datasets to evaluate the method against existing AFA techniques.

### Strengths


### Weaknesses

### Questions

**Strengths:**

- The use of SHAP values for feature ranking supervision in AFA is novel, bringing interpretability-focused methods into the AFA setting.
- The two-stage training strategy seems interesting. It combines imitation learning and on-policy adaptation to aligns feature acquisition with SHAP rankings, improving the model’s robustness.
- The authors provide a farily clear description of the model's structure and training pipeline, which facilitates reproducibility.

**Weaknesses:**

- Reliance on SHAP values for ranking: The effectiveness of SHAP values as a basis for feature acquisition is questionable. Previous works, such as "Marginal Contribution Feature Importance - an Axiomatic Approach for Explaining Data" (Amnon Catav, et al. ICML 2021), have pointed out that SHAP can dilute the importance of redundant features, while "Problems with Shapley-value-based explanations as feature importance measures" (IE Kumar, et al. ICML 2020) highlights how SHAP’s equal distribution of influence may not suit non-additive models. These limitations challenge the suitability of SHAP-based ranking for supervising feature acquisition policies.
- Baseline: The evaluation lacks an essential baseline where features are acquired based on a global feature importance ranking. Specifically, determining a global ranking from the training dataset and using this static ranking to acquire features for each test instance would offer a valuable comparison, assessing the advantage of instance-specific rankings in this AFA context.
- Dataset selection: The experiments focus purely on image classification datasets, which are not the most relevant to AFA. In practical applications like healthcare or finance, where feature acquisition is costly, a dynamic approach to selecting features would be more impactful. Without evaluations on these types of datasets, the real-world utility of the proposed method remains unclear.

**Questions:**

- How might this model perform on datasets more representative of typical AFA applications, such as those in medical diagnostics, where feature acquisition incurs genuine costs? In such scenarios, I believe that acquisition based on global feature ranking is a strong baseline.
- To increase the evaluation's persuasiveness, consider conducting experiments on various tabular datasets.
- How does the model’s SHAP-based feature selection strategy compare with alternative methods, such as conditional mutual information, especially in scenarios with highly correlated features?

---

> ### Author Response · Authors · 2024-11-26
> **Response to grQh's comments**
>
> Response to W1: We appreciate the reviewer’s insightful comment and agree that SHAP values may have some limitations, as highlighted in the referenced works. However, we would like to emphasize the following points: (a) Our proposed method is agnostic to the specific ranking mechanism used. Any improvement in feature importance ranking, whether through SHAP or alternative methods, directly enhances the performance of our framework. Thus, our approach can easily adapt to advancements in local explanation techniques. (b) While local explanation methods like SHAP are one of the many approaches to determine feature importance, our experiments demonstrate that they can effectively generate instance-specific feature ranking orders. This is particularly useful in guiding feature acquisition policies. (c) We selected SHAP due to its widespread adoption and robust theoretical foundation, making it a natural baseline for comparison. Furthermore, approximation techniques like FastSHAP significantly enhance its computational feasibility, making it practical for our use case. (d) We acknowledge the limitations of SHAP highlighted in the cited works. However, our method is designed to seamlessly incorporate more accurate or specialized ranking techniques as they become available. This ensures that the framework can continuously improve alongside advancements in feature importance estimation. To further support our framework, we performed additional experiments using other explainability methods and showed encouraging results. Please see the common response. In summary, while SHAP has inherent limitations, its practicality and instance-specific utility make it a reasonable choice for our work. As better local explanation methods are developed, our approach will readily integrate these improvements to deliver enhanced results.
>
> Response to W2: We appreciate the reviewer’s suggestion and agree that a comparison to a baseline using global feature importance rankings provides valuable insights. To address this, we included CAE (concrete autoencoders) as a baseline in our evaluation. CAE identifies discrete features that are representative of the training data, effectively providing a global feature ranking. Prior work has also established CAE as a standard baseline for evaluating acquisition policies based on global feature rankings. In our experiments, we compared our instance-specific ranking approach against CAE to demonstrate the added value of tailoring feature acquisition to individual test instances. Our results indicated that acquisition based on instance-specific rankings achieved superior performance compared to CAE. We believe this comparison addresses the concern, as it evaluates the effectiveness of global rankings while highlighting the advantages of our instance-specific approach.
>
> Q3: We compared our method against the GDFS and DIME methods, which are CMI-based. Please see the common response and submission for the results.
>
> Response to W3, Q1, & Q2: As per the reviewer's suggestion, we evaluated the model performance on other tabular medical datasets as well as explored other feature ranking methods. Please see the common response.

---

### Author Response · Authors · 2024-11-25
**Common response**

We acknowledge the valuable suggestions from the reviewers and have made several improvements to the manuscript in response. Specifically, we addressed the following concerns:

1) We expanded our evaluation to include tabular medical datasets.

- We used a dataset based on a study published in Hepatology (https://archive.ics.uci.edu/dataset/878/cirrhosis+patient+survival+prediction+dataset-1), which focused on predicting survival on individual patients with primary biliary cirrhosis (CPS). The dataset contained 312 patients from Mayo Clinic with tabular features, and the goal was to perform a 3-label classification task (0 = D (death), 1 = C (censored), 2 = CL (censored due to liver transplantation)). On this dataset, our model achieved an accuracy of $67.12\\%$ averaged over 8 features. In comparison, the GDFS method [1] achieved an accuracy of $65.57\\%$, and the DIME method [2] achieved an accuracy of $66.91\\%$.

- We explored another dataset based on a study published in the New England Journal of Medicine (https://archive.ics.uci.edu/dataset/890/aids+clinical+trials+group+study+175), which is AIDS Clinical Trials Group Study (CTGS). The dataset contained healthcare statistics and categorical information about patients who have been diagnosed with AIDS, totaling 23 features. The task is to predict whether or not each patient died within a certain window of time or not, i.e., a binary classification task. For this problem, our model achieved an AUROC of $0.9167$ averaged over 20 features. In comparison, the GDFS method achieved an AUROC of $0.9142$ and the DIME method achieved a score of $0.8947$.

- We leveraged another dataset that contained detailed health information for 1,659 patients diagnosed with chronic kidney disease (CKD) (https://www.kaggle.com/datasets/rabieelkharoua/chronic-kidney-disease-dataset-analysis). The dataset comprised of 50 features including demographic details, clinical measurements, medication usage, symptoms, quality of life scores, environmental exposures, and health behaviors. The goal was to predict the presence of CKD, which is a binary classification task. Our model achieved an AUROC of $0.8473$ averaged over 20 features, whereas, the GDFS method achieved $0.8060$ and the DIME method achieved $0.7615$.

- Finally, on the Metabric dataset, our model achieved $69.90\\%$ accuracy when averaged over 20 features. In comparison, DIME achieved $71.45\\%$, and the GDFS method achieved $70.11\\%$.

For more info such as performance on these datasets for the Oracle setting, please see the first table below.

2) We incorporated concrete autoencoders (CAE) as a baseline [3], which is a static global feature ranking approach. This comparison emphasizes the advantages of our instance-specific ranking framework over global ranking methods.
The CAE baseline achieved accuracies of $57.03\\%$ and $59.59\\%$ on the CPS and the Metabric datasets, respectively. Also, the CAE baseline achieved AUROC scores of $0.6018$ and $0.8653$ on the CKD and CTGS datasets, respectively. See above for the comparison with our method.

3) As per the reviewer suggestions, we performed an ablation study to evaluate the effectiveness of using decision transformer versus another architecture, and compared the model results. On CIFAR-10 (CIFAR-100) dataset, we found that our model with decision transformer achieved an accuracy of $78.61\\%$ ($47.06\\%$) averaged over 20 image patches. When the decision transformer was replaced with a ResNet block, the model achieved an accuracy of $76.83\\%$ ($46.70\\%$). On the BloodMNIST dataset, we found that our model with decision transformer achieved an accuracy of $87.93\\%$ averaged over 20 image patches. When the decision transformer was replaced with a CNN block, the model achieved an accuracy of $78.23\\%$. These results demonstrate the advantage of using a decision transformer as the policy network while also highlighting that our method performs reasonably well with other architectures as the policy network.

4) We leveraged LIME to demonstrate the flexibility of our approach in leveraging other feature ranking methods. On the CPS dataset, our model achieved an accuracy of $66.21\\%$ when LIME-based ranking was used. See above for comparison with SHAP-based ranking. Furthermore, we observed that the model performance was relatively unchanged when different SHAP-based feature ranking methods were used. For example, KernelSHAP and TreeSHAP resulted in accuracies of $66.06\\%$ and $65.91\\%$, respectively. For the results on the other datasets, please see the second table below.

---

> ### Author Response · Authors · 2024-11-26
> **Common response - continued**
>
> The following table presents our model’s performance compared to other methods on the Metabric, CPS, AIDS, and CKD datasets. Performance metrics were calculated by averaging scores across 20 features, except on the CPS dataset, where the scores were averaged over all 8 features. Please see Figure 2 in the revised manuscript for non-averaged performance comparisons across different numbers of features.
>
> |  | Metabric (6 classes) | CPS (3 classes)  | CTGS (2 classes)  | CKD (2 classes)  |
> |----|----------------------------------------|--------------------------|---------------------------|--------|
> | Oracle | 91.97% | 72.59% | 0.9208 | 0.8862 |
> | Our method | 69.90% | 67.12% | 0.9167 | 0.8473 |
> | DIME | 71.45% | 66.91% | 0.8947 | 0.7615 |
> | GDFS | 70.11% | 65.57% | 0.9142 | 0.8060 |
> | CAE | 59.59% | 57.03% | 0.8653 | 0.6018 |
>
> We also tested our method performance using various feature ranking approaches on the medical tabular datasets. Specifically we used four local explanation methods: TreeSHAP [4], LIME [5], KernelSHAP [6], and Sampling (IME) [7]. For the tabular datasets, our default SHAP calculation method is TreeSHAP. The following table presents the averaged results as the previous table. For the details, please see the revised manuscript.
>
> |  | Metabric (6 classes) | CPS (3 classes)  | CTGS (2 classes)  | CKD (2 classes)  |
> |----------------------|--------------|-----------|-----------|-----------|
> | TreeSHAP         | 69.90%       | 67.12%    | 0.9167    | 0.8473    |
> | LIME             | 69.04%       | 66.21%    | 0.9125    | 0.8122    |
> | KernelSHAP       | 70.01%       | 66.06%    | 0.9146    | 0.8353    |
> | Sampling (IME)   | 69.72%       | 65.91%    | 0.9160    | 0.8152    |
>
>
>
> References:
>
> [1] Covert et. al.,, “Learning to maximize mutual information for dynamic feature selection,” vol. 202 of Proceedings of Machine Learning Research, pp. 6424–6447, PMLR, 23–29 Jul 2023.
>
> [2] Gadgil et. al., “Estimating conditional mutual information for dynamic feature selection,” in The
> Twelfth International Conference on Learning Representations, DOI: 10.48550/arXiv.2306.03301, 2024.
>
> [3] Balın et. al., “Concrete autoencoders: Differentiable feature selection and reconstruction,” in vol. 97 of Proceedings of Machine Learning Research, pp. 444–453, PMLR, 09–15 Jun 2019.
>
> [4] Scott M Lundberg, Gabriel Erion, Hugh Chen, Alex DeGrave, Jordan M Prutkin, Bala Nair, Ronit Katz, Jonathan Himmelfarb, Nisha Bansal, and Su-In Lee. From local explanations to global understanding with explainable ai for trees. Nature machine intelligence, 2(1):56–67, 2020.
>
> [5] Marco Tulio Ribeiro, Sameer Singh, and Carlos Guestrin. "Why should i trust you?": Explaining the predictions of any classifier. In Proceedings of the 22nd ACM SIGKDD International Conference on Knowledge Discovery and Data Mining, KDD ’16, pp. 1135–1144, New York, NY, USA, 2016.
>
> [6] Scott M Lundberg and Su-In Lee. A unified approach to interpreting model predictions. In I. Guyon, U. Von Luxburg, S. Bengio, H. Wallach, R. Fergus, S. Vishwanathan, and R. Garnett (eds.), Advances in Neural Information Processing Systems, volume 30. Curran Associates, Inc., 2017.
>
> [7] Erik Štrumbelj and Igor Kononenko. An efficient explanation of individual classifications using game theory. Journal of Machine Learning Research, 11(1):1–18, 2010.

---

### Note · Authors · 2025-02-11

I have read and agree with the venue's withdrawal policy on behalf of myself and my co-authors.

---

### Meta-Review · Area_Chair_pnBR · 2024-12-21

**Metareview:**

This paper proposes a novel active feature acquisition framework that leverages SHAP to generate instance-specific feature importance rankings. By reframing AFA as a feature prediction task, the authors employ a decision transformer-based policy network and a two-stage training strategy (imitation learning followed by on-policy adaptation). The proposed method demonstrates strong empirical performance across several datasets, outperforming state-of-the-art AFA techniques in terms of predictive accuracy and acquisition efficiency.

**Strengths:**
* The use of SHAP values to guide instance-specific feature acquisition is a novel (bridging interpretability and AFA)
* The two-stage training strategy shows promising results.
* Empirical results demonstrate that the proposed method outperforms existing AFA techniques in terms of accuracy and acquisition efficiency.
* The authors provided additional experiments on tabular medical datasets in response to reviewer feedback, addressing concerns about dataset relevance.

**Weaknesses:**
* Reliance on SHAP for feature ranking remains controversial due to its known limitations, particularly with redundant features or non-additive models.
* Most experiments are conducted on image datasets (limiting the real-world applicability of the method)
* It does not dynamically recalculate feature importance, potentially missing dependencies between already acquired and remaining features.
* Lack of theoretical justification for why SHAP-based rankings should lead to optimal feature acquisition

In summary, the paper presents an interesting and novel approach, but concerns about real-world applicability, reliance on SHAP, and lack of theoretical grounding are significant. The additional experiments help strengthen the paper but did not fully resolve the core issues raised.

**Additional Comments On Reviewer Discussion:**

The discussion emphasized concerns about the static SHAP-based ranking and the lack of dynamic, context-aware feature evaluation. Despite additional experiments, reviewers found these limitations critical.

---

### Decision · Program_Chairs · 2025-01-22

Reject